# A Wide-Band Low-Profile Antenna for a High-Integration Phased Array System

**DOI:** 10.3390/s24113295

**Published:** 2024-05-22

**Authors:** Haipeng Liu, Juan Liu, Jin Huang, Chunhui Han, Bo Chen, Yuhe Liu, Yujie Xiang

**Affiliations:** 1School of Mechano-Electronic Engineering, Xidian University, Xi’an 710071, China; 2Beijing Institute of Remote Sensing Equipment, Beijing 100039, China

**Keywords:** tightly, antenna, modular, phased array

## Abstract

In this paper, a wide-band, low-profile antenna is presented for a high-integration phased array system. The proposed antenna, implemented using a tightly coupled array, operates over roughly the X-K frequency band and is performant at 8 GHz–18.5 GHz. The antenna can scan to ±60 degrees in both the E- and H-planes. Compared to previous tightly coupled antennas with smaller element spacing, the antenna in this paper reaches 9.4 mm, which corresponds to 0.58 λ of high frequency, suitable for engineering application conditions in production. The antenna can be soldered to BGA T/R chips in this space. Additionally, to facilitate flexible assembly for large arrays, the antenna is manufactured modularly using four elements and its parasitic radiation is analyzed. Then, a method for repressing parasitic radiation is presented. Finally, the antenna is fabricated and measured in a microwave chamber, exhibiting an excellent pattern and scanning radiation. The measured performance agrees with the full-wave finite array simulations.

## 1. Introduction

As communication and radar devices develop, more intensive requirements are proposed in terms of antenna performance, such as a wider bandwidth and a larger scanning angle. A tightly coupled array is a broadband antenna with good scanning performance. It was first developed based on the continuous current sheet model [1], whose impedance did not change with frequency. Later, it was further studied by Professor Munk [2] and subsequently underwent additional scholarly development. S. S. Holland designed a tightly coupled antenna array covering 7–21 GHz, with strong scanning but low mechanical strength [3]. Professor Volakis implemented a seven-time-band tightly coupled antenna with overlapping dipoles, reaching a scanning angle of 45 degrees [4,5,6]. Furthermore, Professor Ersin Yetisir used a frequency-selective surface for the matching layer and realized six-time-band broadband, reaching 70 degrees in the E-plane and 65 degrees in the H-plane [7]. Tightly coupled antennas usually realize a larger bandwidth with a small array space [8]. When the array space becomes larger, the performance deteriorates sharply. The array space of a tightly coupled antenna is generally about 0.4–0.5 times the wavelength of its highest frequency. As the antenna with the highest frequency is 18 GHz, its array space is about 7.5 mm. This makes it difficult to engineer in a practical manner, especially for high-density integrated antennas, where the antenna unit is welded directly to a TR chip and the array space is limited by the chip size. At the same time, antennas also need to be modular for convenient production and flexible splicing. Therefore, it is necessary to study a tightly coupled antenna array which is suitable for a high-density integrated phased array system.

In this paper, we will realize a tightly coupled array with a larger space which can be used for a highdensity integration phased array system. An 8 × 8 antenna array with a broad band of 8–18.5 GHz is designed, meeting the requirement for a 60-degree scanning angle in the E-plane and the H-plane. The design considers engineering requirements, including the size matching of antennas and chips, increasing the success rate of antenna production through small array production, and flexible assembly. The difference between the antenna in this paper and conventional tightly coupled antennas is that this antenna is welded with a TR chip, so the array space needs to accommodate the welding technology, requiring it to be much larger than the conventional array space. Engineering applications demand a larger lattice space, which also benefits heat radiation. At the same time, when the tightly coupled antenna is minimally modularized, the continuity of its current will be broken, which will lead to a decline in its performance. Parasitic radiation will appear and seriously interfere with the directional pattern. In light of all of this, this paper designed and optimized a tightly coupled antenna with a large array lattice space for practical engineering applications. Equally, a solution to the parasitic radiation is analyzed and verified. Finally, an 8 × 8 antenna array with 9.4 mm of space is realized and applied to a radar or communication system.

## 2. Antenna Element Design and Analysis

### 2.1. Antenna Element Design

The extensively studied basic structure of a tightly coupled antenna is shown in Figure 1 [9]. Evolved from the infinite current sheet model, the antenna utilizes the mutual coupling effect between elements to form a continuous current distribution. Its radiation impedance is only related to the scanning angle and is independent of frequency, achieving a wider operating bandwidth of the antenna. In this paper, this skeleton is improved upon and optimized. Unlike before, a large-sized design is adopted and a metal strip is loaded. The size of the radiation layer, feeder layer and coupled patch are adjusted, and a strip line feeder is designed under the ground. Metallic strips are loaded onto the radiation layer on the matching layer to improve the VSWR [10]. Figure 1a is a three-dimensional structure model of the antenna, and Figure 1b is a two-dimensional structural model of the antenna from the front view. It consists of a matching layer, a radiation layer, a ground layer, a feeder layer, and a strip layer. The element feeding ports are soldered to the chip and the RF signal is transported spherically and coaxially. The impedance of the antenna port and the chip port is 50 Ω. The E-plane and the H-plane are noted as the *xz*-plane and the *yz*-plane.

The key parameters are shown in Figure 2, and the element dimensions are displayed in Table 1. The antenna has a low profile, with a height of 5.4 mm.

To satisfy the engineering requirements, the antenna’s lattice space is 9.4 mm × 9.4 mm. According to the array scanning theory, when there is no scanning lobe in the visible space, the array can scan to 60 degrees at 8–17 GHz and 50 degrees at 17–18.5 GHz. The formula is presented as follows:(1)$$d<\frac{\lambda }{1+sin\theta }$$
where *d* is the distance between the adjacent elements, λ is the wavelength in the free space, and *θ* is the scanning angle.

The antenna is simulated using high-frequency software HFSS 2021. Periodic boundary conditions are used to simulate an infinite array environment. From the obtained results, the antenna bandwidth is 8–18.5 GHz. The analysis suggests the length of the radiation patch affects its high-frequency performance and the air cavity size, and the thickness between the coupled patches affects its low-frequency performance. The matching layer design also affects the scanning and frequency performance. The final model parameters are shown in Table 1.

Figure 3 describes the simulated VSWR in the infinite environment, including different scanning angles in the E-plane and the H-plane, which are 30 degrees and 60 degrees. For the integrated phased array antenna, a good VSWR is important because of the integration with the TR chip, for which the power amplifier requires a matched load [11]. It can be shown that the VSWR is less than 2.3 across the frequency band of 8–18.5 GHz with no scanning. When scanning to 30° in the H-plane, the active VSWR is less than 2.7. Meanwhile, when the scanning angle is 60°, the active VSWR is less than 4.3 across 8–17 GHz. A grating lobe will appear if the frequency is more than 17 GHz when the scanning angle is 60°. In the E-plane, the active VSWR is less than 3.1 when scanning to 30°, and the active VSWR is less than 5 when scanning to 60°. Therefore, the VSWR is sufficient for the phased array application.

The 8 × 8 array is simulated in the full-wave analysis software HFSS with the port excitation synthesized [12], and dummy elements are loaded onto the side of the model to reduce the problem of current reflection caused by the truncation of the array [13]. The excitation distribution of the array is uniform. Figure 4 and Figure 5 show the performance of the normalized pattern simulated using the high-frequency software, including several frequencies in the E-plane and H-plane scanning patterns. The frequencies are 8, 10, 12, 14, 17, and 18.5 GHz, with a grating lobe appearing at a frequency over 17 GHz when scanning to 60°. All of the patterns have good scanning performance.

The realized gain and efficiency were analyzed, and as seen in Figure 6, the realized gain is 15 dB in 8 GHz and 23.9 dB in 18.5 GHz, and the averaged efficiency is computed to be about 80% while taking into account mismatch losses. The efficiency decreases rapidly from 8.5 GHz, which is due to the deterioration of the VSWR.

### 2.2. Several Engineering Problems including Production and Welding

The antenna designed in this paper is applicable in a high-density integration phased array system. However, welding the antenna element with a TR chip will result in several engineering problems.

Firstly, conventional tightly coupled arrays are produced in one go. As shown in reference [14], the antenna array ensures continuity on the E-plane. In this paper, when welding a chip with the antenna with the requirement of flexible assembly, the antenna is produced and welded according to a 2 × 2 model, and the size of each sub-array is 18.6 × 18.6 mm, which is considered the tolerance of the process [15]. When numerous sub-arrays are assembled into a large array, there is a gap of 0.2 mm between the sub-arrays, which is fatal to the tightly coupled array antenna and its working performance. The testing results show that when the antenna array has a large scanning angle, its pattern has a remote side lobe, which seriously affects the system’s use.

Secondly, each sub-array is welded with TR chips, the assembly model of which is shown in Figure 7. Because the chip feeding position does not exactly match the antenna element port, the antenna has an additional feeding transition layer to adjust the antenna feeding position to the chip feeding port. During PCB processing, the solder pad is plated after plugging the hole with resin, causing the center of the solder pad to sag. In order to flatten the antenna planting ball pad, the through hole of the antenna is plugged with copper paste. In addition, the chip consumes a lot of heat due to its high power, so the antenna welding surface is formed of thick copper, which can alleviate heat accumulation in the chip.

### 2.3. Analysis and Verification of Parasitic Radiation Suppression

An unwanted antenna lobe arising in an engineering application is modeled, as shown in Figure 8, simulated, and analyzed. A relatively high side lobe is generated in the far region during antenna scanning in the E-plane, and the position of the side lobe remains unchanged when changing the scanning angle. According to the simulation of the antenna electric field in 15 GHz, a strong field is distributed in the ground gap of the array splicing. The continuity of the current between the sub-arrays is destroyed, and the field distribution in the gap will change when the frequency is changed. The E-field is perpendicular to the long edge of the gap, as seen for slot antennas [16,17]. It is different from a normal field, which is vertical to the ground. Thus, parasitic radiation is generated.

In order to suppress the parasitic radiation generated by the gap, a solder strip is loaded onto the gap to connect the sub-arrays, a model for which is depicted in Figure 9a. The strip is also easily soldered to the ground plane of the antenna. The analyzed field distribution is shown in Figure 9b. It is can be seen that the field has changed, as it is vertical to the ground and different from the non-loaded antenna.

An antenna loaded with a solder strip is also simulated. As an example, at 15 GHz and 16.5 GHz, parasitic radiation is evidently suppressed, as seen in Figure 10. The simulation of the model loaded with a solder strip shows that parasitic radiation is effectively suppressed.

As a result, according to the loading of the solder strip, the characteristics of the E-plane pattern are significantly improved. The high side lobe is suppressed; therefore, sub-arrays can successfully be used to make a large-scale array.

## 3. Manufacturing of an Antenna

Figure 11 displays a sub-array manufactured using PCB technology. The assembly of four sub-arrays is shown in Figure 11a, and the 0.2 mm gap is evident. The feeding port position is shown in Figure 11b, and the solder pad has a 250 um radius. The antenna loaded with a solder strip is shown in Figure 11c, and the gap is bridged using the solder strip. The length of solder strip is about 3 mm. The chip and the antenna are welded together using planting balls, and the actual object is tightly integrated after assembly.

The antenna is finally integrated into a 400-element radar system, experimentally verified to have good working performance, with successful antenna production, welding, and integration of 9600 array elements.

## 4. Experimental Results

The above design was experimentally tested, verifying the pattern and the VSWR of an 8×8 array. The VSWR of the element in the array was tested by an Agilent network analyzer and three-jaw probe, before the antenna was integrated with the chip. The probe was used to test antenna port. One claw probe was connected to the feeding pad, while the other two claws were connected to the ground. The results are shown in Figure 12. It can be seen that the VSWR is less than three. The difference between the simulated and measured results is caused by production tolerance.

To test and verify its scanning patterns, the 8 × 8 antenna array was integrated into the active phased array system, and the amplitude and phase of the chip were controlled using a computer. When we performed the antenna scanning, we used a computer to calculate the phase required for antenna unit, and control the chip to realize these phases. The assembled antenna array was placed in a microwave darkroom to measure its E-plane and H-plane patterns at different frequencies. The measured normalized patterns are shown in Figure 13 and Figure 14.

The measured normalized scanning radiation patterns in the E-plane and the H-plane at frequencies of 8 GHz, 10 GHz, 12 GHz, 14 GHz, 17 GHz, and 18.5 GHz are shown in Figure 13 and Figure 14, with scanning angles from −60° to 60°, and the measured radiation patterns are performant. At a frequency of 18.5 GHz, these patterns will deteriorate when scanning to 60° due to a grating lobe, in accordance with the simulation results.

To show the agreement between the simulated and measured patterns clearly, several frequencies were selected for comparison. In Figure 15, the patterns are almost consistent with the simulated patterns. At a small scanning angle, the directional patterns agree, but there is a minor difference between the simulated and measured patterns at a large scanning angle. This is due to the inconsistent coupling characteristics caused by processing and assembly. With the antenna comprising sub-arrays, although the gaps are connected with solder strips, it still differs from the simulation model slightly.

The results comparing the proposed antenna array and other similar arrays are shown in Table 2. The proposed array has a large lattice space and is modularized using four elements, which is convenient for engineering applications.

Finally, the processed array is determined to have good performance for successful use in electronic or communication systems. There is no blind scanning from 8 GHz to 18.5 GHz using the antenna array. Therefore, we have ascertained the appropriate methods for the design and production of a large-lattice broadband antenna array suitable for high-density integrated systems. The proposed antenna is used in a phased array system, which has a low profile and high integration, including RF circuits, control circuits, power supply circuits, and heat dissipation. It has also been verified in a system.

## 5. Conclusions

A wide-band antenna array for a high-density integration phased array system is designed in this paper. The antenna has good E-plane and H-plane pattern characteristics in the frequency range of 8–18.5 GHz. In order to match the engineering requirements and integrate with the RF chips, the antenna is different from the previous coupled antennas which had a small lattice space. The array adopts the idea of large lattice space and modularization, and solves the problem of parasitic radiation caused by the tightly modular assembly. The antenna array is measured and verified, and the results are consistent with the simulated results. The antenna can be used in broadband low-profile radar or communication systems.

## Figures and Tables

**Figure 1 sensors-24-03295-f001:**
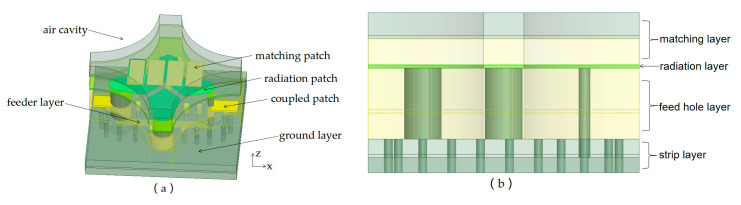
Geometry of the proposed tightly coupled antenna element. (**a**) Three-dimensional view. (**b**) Front view.

**Figure 2 sensors-24-03295-f002:**
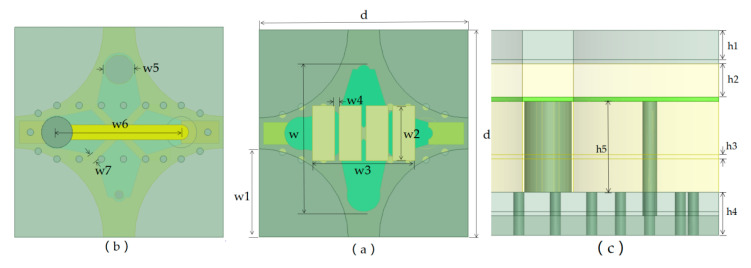
Parameters of the proposed tightly coupled antenna element. (**a**) Bottom view. (**b**) Top view. (**c**) Side view.

**Figure 3 sensors-24-03295-f003:**
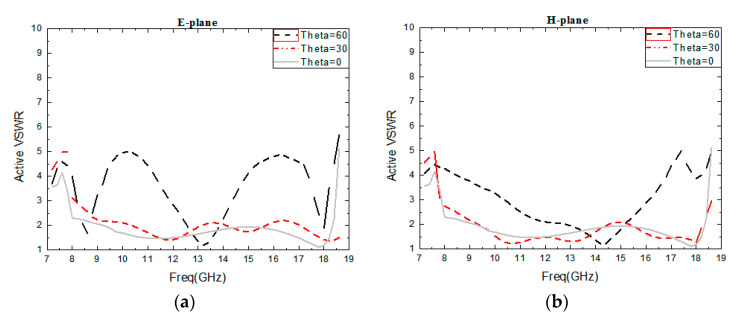
Simulated infinite active VSWR of the tightly coupled antenna at 0°, 30°, and 60° scanning in the E- and H-planes: (**a**) H-plane and (**b**) E-plane.

**Figure 4 sensors-24-03295-f004:**
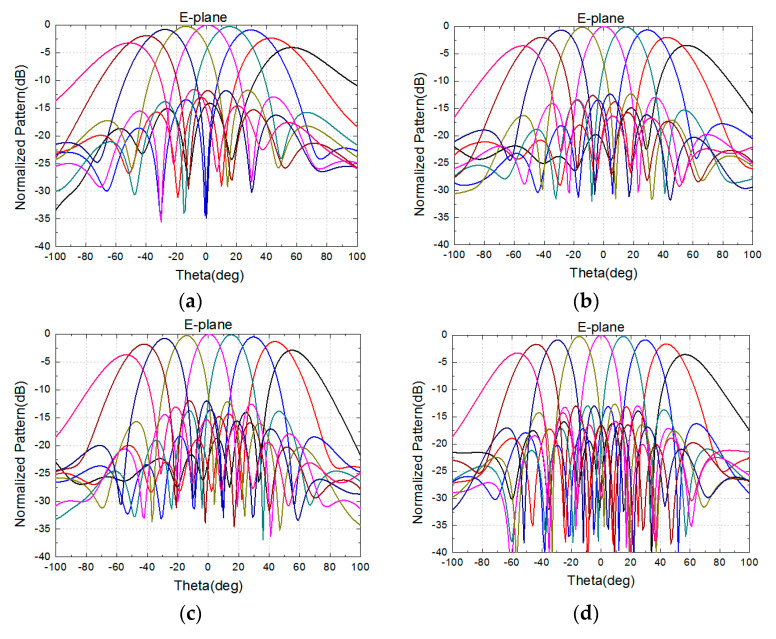
Simulated normalized radiation patterns of 8 × 8 array in the E-plane. (**a**) 8 GHz. (**b**) 10 GHz. (**c**) 12 GHz. (**d**) 14 GHz. (**e**) 17 GHz. (**f**) 18.5 GHz.

**Figure 5 sensors-24-03295-f005:**
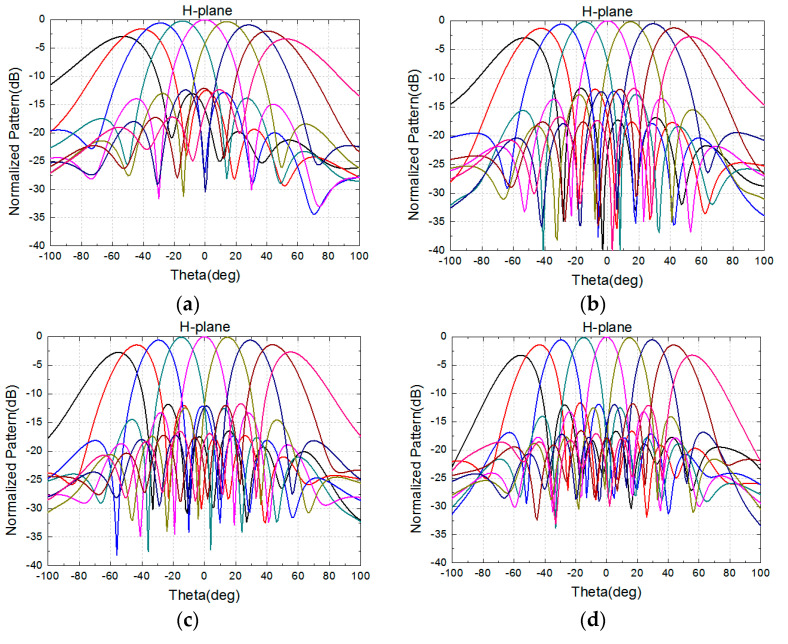
Simulated normalized radiation pattern of 8 × 8 array in the H-plane. (**a**) 8 GHz. (**b**) 10 GHz. (**c**) 12 GHz. (**d**) 14 GHz. (**e**) 17 GHz. (**f**) 18.5 GHz.

**Figure 6 sensors-24-03295-f006:**
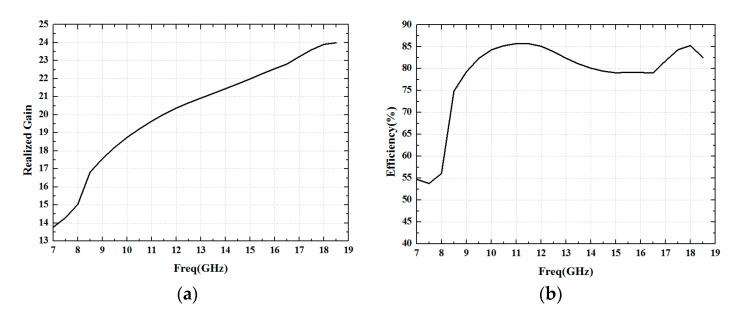
Simulated realized gain and efficiency for 0 deg beam steering: (**a**) realized gain and (**b**) efficiency.

**Figure 7 sensors-24-03295-f007:**
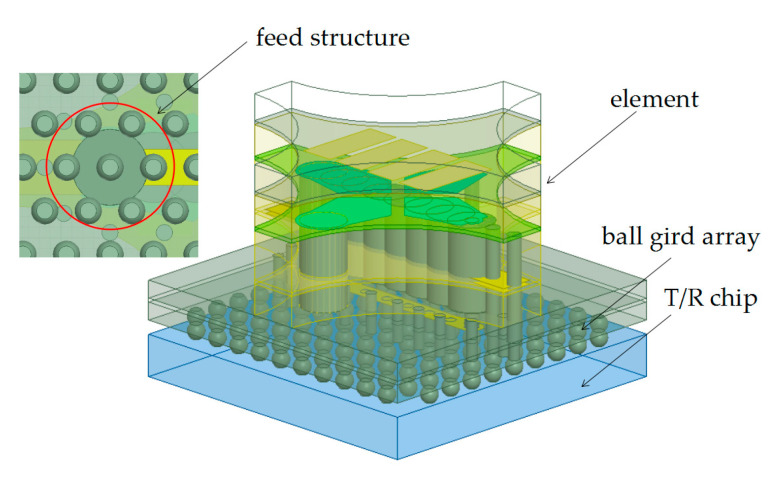
Assembly model of the element with T/R chip.

**Figure 8 sensors-24-03295-f008:**
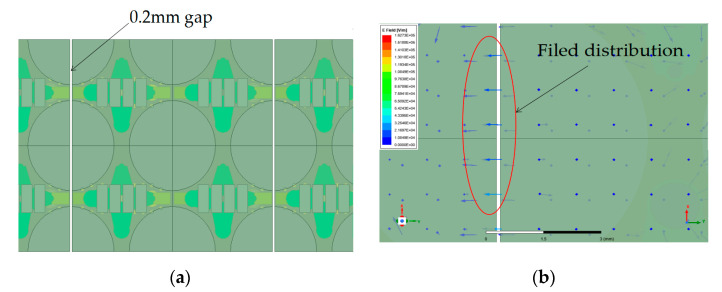
Simulation of non-loaded antenna lobe in 15 GHz. (**a**) Simulation model with gap of engineering application. (**b**) Field distribution in the ground gap of the array splicing.

**Figure 9 sensors-24-03295-f009:**
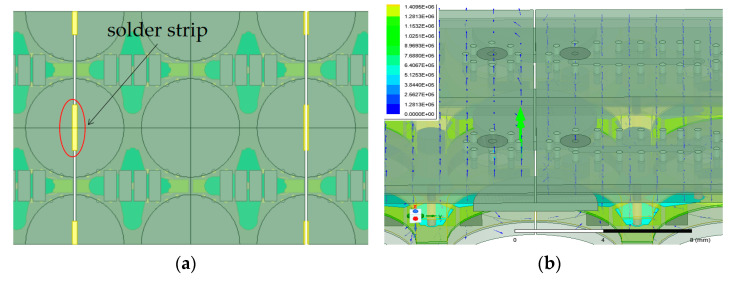
Simulation of loaded antenna lobe in 15 GHz. (**a**) Simulation model of the loaded solder strip. (**b**) Field distribution in the ground gap of the loaded solder strip.

**Figure 10 sensors-24-03295-f010:**
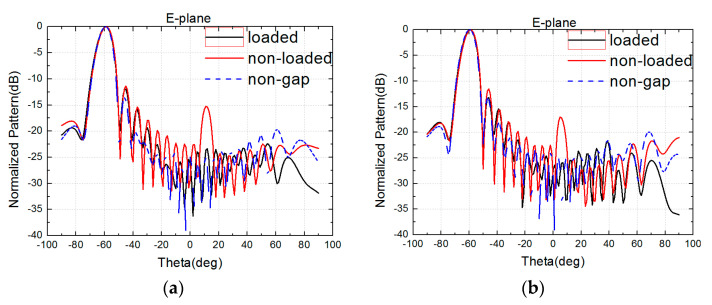
Field distribution in the ground gap with copper loading. (**a**) 15 GHz. (**b**) 16.5 GHz.

**Figure 11 sensors-24-03295-f011:**
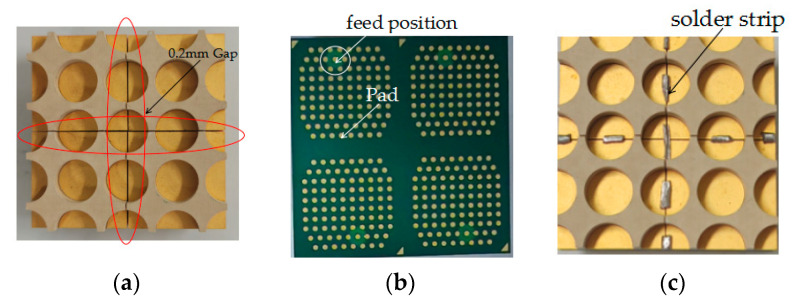
Sub-arrays manufactured using PCB technology. (**a**) Sub-array with gap. (**b**) Feed position and pad. (**c**) The antenna loaded with a solder strip.

**Figure 12 sensors-24-03295-f012:**
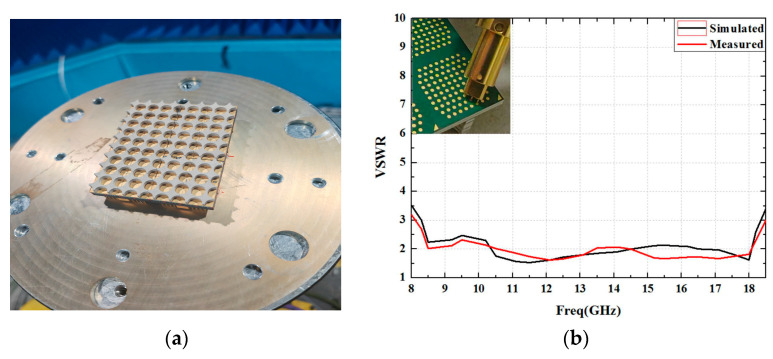
The simulated and measured VSWRs of the element in the array. (**a**) Array prototype. (**b**) Measured and simulated results.

**Figure 13 sensors-24-03295-f013:**
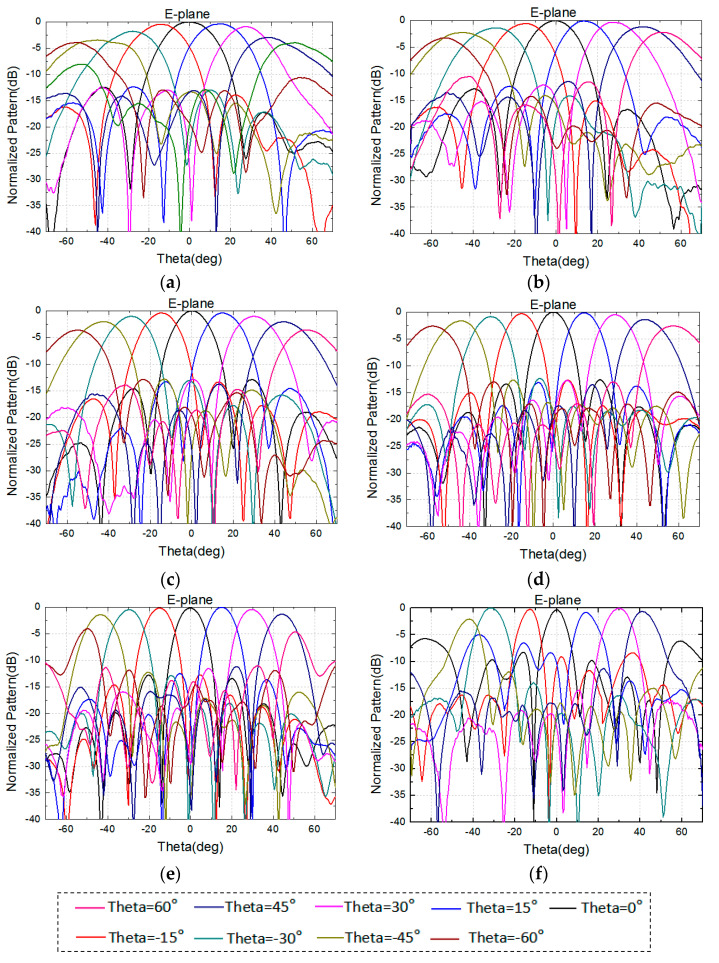
Measured normalized radiation patterns of 8 × 8 array in the E-plane. (**a**) 8 GHz. (**b**) 10 GHz. (**c**) 12 GHz. (**d**) 14 GHz. (**e**) 17 GHz. (**f**) 18.5 GHz.

**Figure 14 sensors-24-03295-f014:**
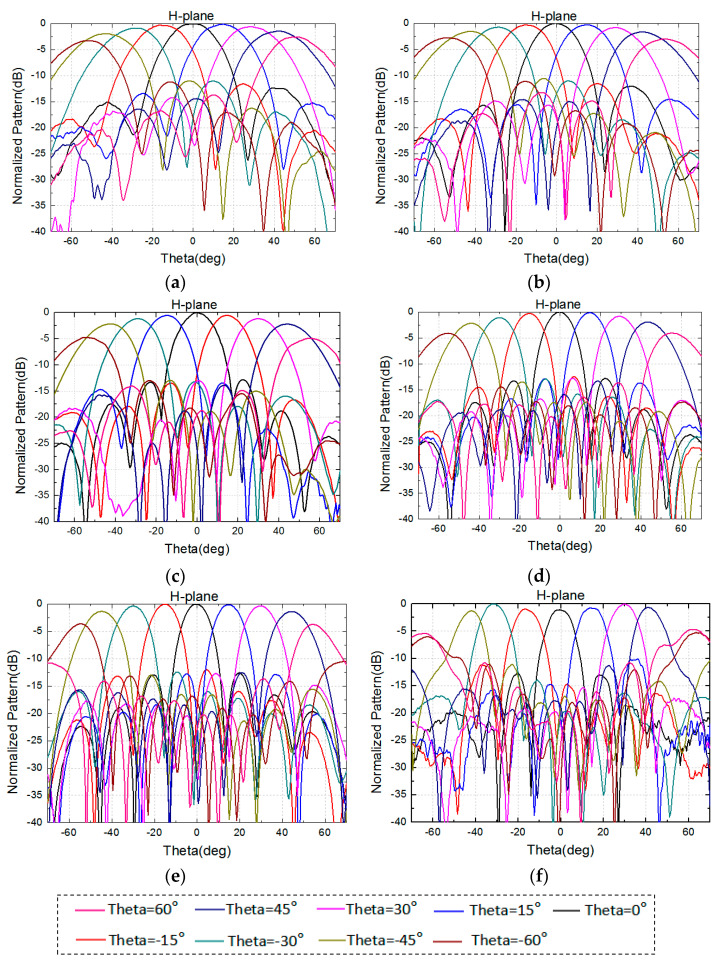
Measured normalized radiation patterns of 8 × 8 array in the H-plane. (**a**) 8 GHz. (**b**) 10 GHz. (**c**) 12 GHz. (**d**) 14 GHz. (**e**) 17 GHz. (**f**) 18.5 GHz.

**Figure 15 sensors-24-03295-f015:**
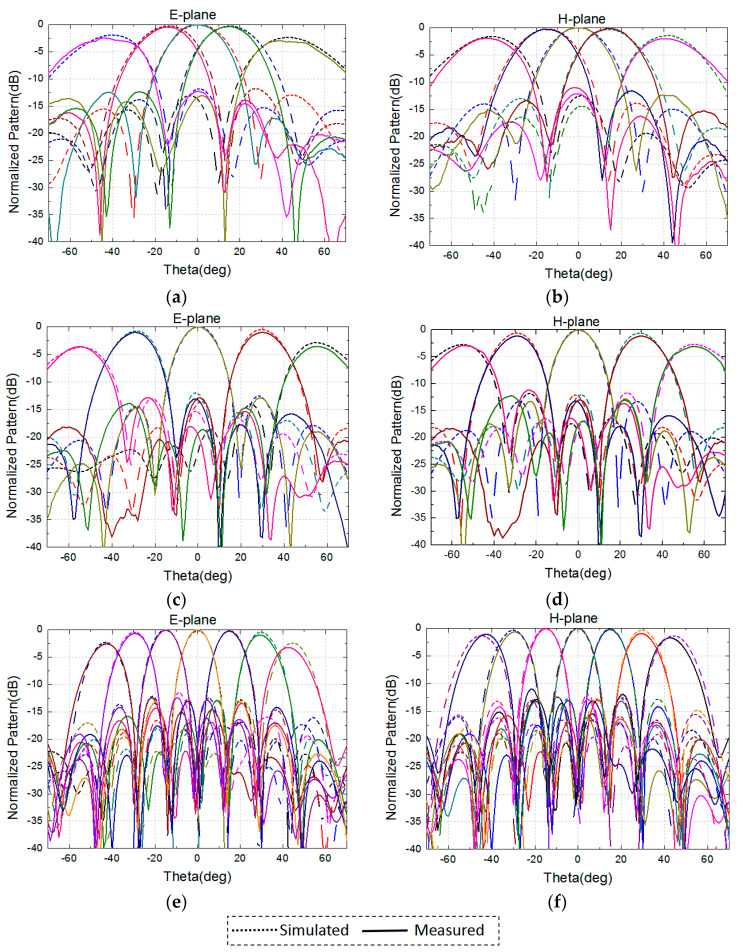
Measured and simulated normalized radiation patterns of 8 × 8 array in the H-plane and E-plane. (**a**) E-plane at 8 GHz. (**b**) H-plane at 8 GHz. (**c**) E-plane at 12 GHz. (**d**) H-plane at 12 GHz. (**e**) E-plane at 17 GHz. (**f**) H-plane at 17 GHz.

**Table 1 sensors-24-03295-t001:** Element dimensions.

Parameter	Value (mm)	Parameter	Value (mm)
h1	0.76	w1	4.00
h2	0.89	w2	2.50
h3	0.11	w3	4.60
h4	1.12	w4	0.20
h5	2.39	w5	1.00
d	9.40	w6	5.60
w	6.50	w7	0.35

**Table 2 sensors-24-03295-t002:** Comparison results for different arrays.

Characteristics	Ref.	Value	Proposed Value
Lattice space	[6]	7.5 mm, 0.46 λ _h_	9.4 mm, 0.56 λ _h_
Production scale	[6]	64	4
Lattice space	[18]	3.75 mm, 0.5 λ _h_	9.4 mm, 0.56 λ _h_
Production scale	[18]	240	4
Lattice space	[19]	6 mm, 0.45 λ _h_	9.4 mm, 0.56 λ _h_
Production scale	[19]	25	4

## Data Availability

Data are contained within the article.

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
