# Peer review of "A Wide-Band Low-Profile Antenna for a High-Integration Phased Array System"

_sensors, 2024, doi:10.3390/s24113295_

Round 1
Reviewer 1 Report
Comments and Suggestions for Authors
1. What is the main problem to be solved in this study?
The authors pursue the idea of maximal densification of the phased antenna arrays
2. What specific gaps in the field does this article address?
The authors seek the trade-off solution among many antenna working parametes, e.g array density, which is related to its overall size, and its working bandwidth.
3. How does it contribute to the subject area compared to other published
papers?
The authors propose a construction which can be integrated to the chip fabrication technoliogy which is currently in use.
4. What specific improvements should the authors consider making to this
method? What further control measures should be considered?
The authors presented consistent and completed study of the problem addressed at the moment.
Any improvements could be considered further, as the general technical and technological proggess goes on.
5. Please describe whether all major questions raised were addressed and what
specific experiments were performed.
Yes, the authors have addressed all these major questions they raised and manufactured adn tested functioning samples of their antenna arrays
and then applied them in practical radar and communication systems.
6. Are the references appropriate?
Yes, they are.
7. Please provide any additional comments on the tables and data and data
quality.
Minor comments.
Line 160, the end. It [is] found that...
The figures numeration. The Fig. 9 follows immediately after Fig. 6. The Fig. numbers 7 and 8 seem to be omitted
Reviewer 2 Report
Comments and Suggestions for Authors
In Section 2, it is unclear from the writing style whether some parts of this antenna design have been previously published. Could the authors please clarify which aspects constitute novel content?
Regarding the discussion in Section 2.1 on the active VSWR, the authors should provide further explanation regarding the interpretation of this metric, particularly its importance for the driving stage of the antenna. Typically, this driving stage consists of a power amplifier that requires a matched load, hence low VSWR. However, data on VSWR can also be useful for testing the amplifier's performance under mismatched loads using techniques such as load-pull, which are now available for broadband excitation. For further reference, see: A. M. Angelotti, et al., "Wideband Active Load–Pull by Device Output Match Compensation Using a Vector Network Analyzer," in IEEE Transactions on Microwave Theory and Techniques, vol. 69, no. 1, pp. 874-886, Jan. 2021. Could the authors elaborate on the phrase "The antenna is simulated with high-frequency software"? Additionally, the statement "The VSWR is enough to be used in the phased array" should be rephrased to "low enough," as the objective is to achieve a low value.
The title "2.2. Several engineering problems" is overly generic. Please specify the particular content of this subsection. It appears that the solder strip may improve the field distribution, but it is unclear what investigation was conducted to reach this conclusion. Please provide further explanation.
In Section 4, is it possible to calculate and provide the graph for the active VSWR under steering conditions?
The authors frequently mention generic "engineering requirements" without stating them in an organized manner. I suggest creating a list of requirements and discussing them at the beginning of the article.
The quality of the figures is not satisfactory. Additionally, the captions of the figures should address each subfigure (this is not done in, for example, Fig. 2). Please note that there are two figures labeled "Figure 2."
Comments on the Quality of English LanguageThe English language of the article is below an acceptable level, and there are several typos. Therefore, I recommend a formal review of English grammar and style.
Reviewer 3 Report
Comments and Suggestions for Authors
1) Overall, the resolution of the pictures is very low. It is recommended to increase the resolution to improve understanding of the paper.
2) Reference [9] was mentioned as the basic structure, but a more detailed explanation of the operating principle would be necessary.
3) It would be nice to mention what improvements have been made compared to reference [10].
4) It is recommended to depict the feed part connected to the chip in more detail in the picture.
5) p3) It is recommended to mention exactly what simulation tool was used.
6) There are some typos. (Ex: Fig. 2 H-plan -> H-plane, Fig. number)
7) In Table 1, it would be nice to mention the specific goal and method of optimization.
8) In Fig. 3a, the left and right sidelobes of the 0-degree beam steering appear asymmetric. Why?
9) It would be recommended to mention the weight distribution of the aperture (Uniform, Taylor, etc.) in Fig. 3.
10) It would be necessary to mention the simulated gain and aperture efficiency (for 0 deg beam steering) in Fig. 3 and 4.
11) It would be recommended to add the simulation frequency in Fig. 5.
12) It would be good to compare the structures without gaps in Fig. 9.
13) It would be nice to mention the optimal length of the solder strip.
14) In Fig. 11, it would be necessary to provide a detailed explanation of the test conditions.
15) It would be nice to add an additional brief explanation and illustration of the applied phased array system.
16) It would be recommended to mention the measurement environment for the radiation pattern.
17) It would be recommended to mention the gain for the measured radiation patterns
Comments on the Quality of English LanguageI think that moderate editing of English language required.
Round 2
Reviewer 2 Report
Comments and Suggestions for Authors
Thanks for replying to the comments. These are satisfactory from a scientific perspective.
Comments on the Quality of English LanguageWhile the authors have slightly improved the manuscript from the first submission, I believe the article still requires proper proofreading in English by some professional service.
Reviewer 3 Report
Comments and Suggestions for Authors
The authors have clearly answered all my previous questions and requests. In my opinion, the paper is now suitable for publication. Congratulations!
Comments on the Quality of English LanguageI think that moderate editing of English language is required.